

# Andrographolide induces protective autophagy and targeting DJ-1 triggers reactive oxygen species-induced cell death in pancreatic cancer

Zhaohong Wang[1,2], Hui Chen[2], Xufan Cai[3], Heqi Bu[4] and Shengzhang Lin[1]

[1] Department of Clinical Medicine, School of Medicine, Hangzhou City University, Hangzhou, China
[2] Department of Hepatobiliary and Pancreatic Surgery, the Second Affiliated Hospital of Wenzhou Medical University, Wenzhou, China
[3] Zhejiang Chinese Medical University, Hanzhou, China
[4] Department of Surgery, Tongde Hospital of Zhejiang Province, Zhejiang Academy of Traditional Chinese Medicine, Hangzhou, China

Corresponding author
Shengzhang Lin, linsz@zucc.edu.cn

## ABSTRACT

**Background**. Andrographolide (Andro), an extract of *Andrographis paniculate* (Burm.f.) Wall. ex Nees (Acanthaceae), possesses diverse biologically active properties. However, the precise mechanisms and effects of Andro on pancreatic cancer (PC) remain unclear.

**Methods**. The cytotoxic potential of Andro and underlying mechanism towards PC cells was investigated through *in vitro* experiments and a xenograft mouse model. PC cells were first subjected to varying concentrations of Andro. The reactive oxygen species (ROS) was assessed using flow cytometry and DCFH-DA staining. The apoptosis rate was detected by flow cytometry. Additionally, western blot was applied to evaluate the expression levels of cleaved-caspase-3, DJ-1, LC3-I, LC3-II, and p62. To further elucidate the involvement of ROS accumulation and autophagy, we employed N-acetylcysteine as a scavenger of ROS and 3-Methyladenine as an inhibitor of autophagy.

**Results**. Andro demonstrated potent anti-proliferative effects on PC cells and induced apoptosis, both *in vitro* and *in vivo*. The cytotoxicity of Andro on PC cells was counteracted by DJ-1 overexpression. The reduction in DJ-1 expression caused by Andro led to ROS accumulation, subsequently inhibiting the growth of PC cells. Furthermore, Andro stimulated cytoprotective autophagy, thus weakening the antitumor effect. Pharmacological blockade of autophagy further enhanced the antitumor efficacy of Andro.

**Conclusion**. Our study indicated that ROS accumulation induced by the DJ-1 reduction played a key role in Andro-mediated PC cell inhibition. Furthermore, the protective autophagy induced by the Andro in PC cells is a mechanism that needs to be addressed in future studies.

## INTRODUCTION

Pancreatic cancer (PC), a highly malignant and invasive form of cancer affecting the digestive system, poses a significant threat to human health with its high mortality rates. By 2030, PC is projected to emerge as the second most common cause of cancer-related mortality (*Xie et al., 2022*). Despite advancements in surgical techniques and chemotherapy for pancreatic ductal adenocarcinoma (PDAC), the 5-year overall survival rate remains dismally low, falling below 7% (*Cascinu et al., 2010*; *Trilla-Fuertes et al., 2022*). Because PDAC lacks early symptoms, most patients have metastasis. For many cases undergoing surgical treatment, the tumor will relapse and metastasize within 1–2 years (*Hariharan, Saied & Kocher, 2008*). The current standard treatment for resectable PDAC involves initial surgery followed by adjuvant chemotherapy. Unfortunately, accompanied by a multitude of adverse effects that profoundly affect the well-being and quality of life of individuals with cancer (*Liu et al., 2021*). Hence, there exists a pressing imperative to advance the development of targeted therapies for PC that effectively mitigate the detrimental effects of conventional treatments and concurrently promote the overall well-being of patients.

Numerous natural plant-derived compounds have demonstrated promising therapeutic potential against tumors and are currently undergoing preclinical and clinical trials (*Naeem et al., 2022*; *Talib et al., 2022*). Among these compounds is andrographolide (Andro) (Fig. 1A), a diterpene lactone from *Andrographis paniculate* (Burm.f.) Wall. ex Nees (Acanthaceae). Andro has been widely used for more than 60 years owing to its extensive therapeutic properties and minimal side effects (*Islam et al., 2018*). Moreover, studies have revealed the anticancer activity of Andro and its derivatives against various cancer cell types, including breast and colon cancer. These compounds exert their effects by impeding the progression of the cell cycle, eliciting apoptosis, and diminishing cellular invasion (*Jada et al., 2008*; *Wong et al., 2014*). However, there is a dearth of research regarding the utilization of Andro in PC treatment, and its mechanism remains to be explored.

DJ-1 is a exquisitely conserved protein, exhibiting a molecular weight of 20 kilodaltons, and it belongs to the protein superfamily known as DJ-1/ThiJ/Pfp I. Amplified expression of DJ-1 has been documented in numerous cancers, including breast cancer (*Scumaci et al., 2020*), melanoma (*Lee et al., 2021*), PC (*Kawate et al., 2015*), astrocytoma (*Haapasalo et al., 2018*) and endometrial cancer (*Benati et al., 2018*). DJ-1 is a protein of extensive ubiquity, intimately implicated in diverse cellular processes, encompassing cellular demise, proliferation, and cellular cycle progression (*Neves et al., 2022*). Its expression can be induced by oxidative stress and it plays a crucial role in cellular response to hypoxia (*Sun & Zheng, 2023*). Notably, DJ-1 confers resistance to diverse apoptotic stimuli in a PTEN-dependent manner, thereby preventing PTEN-induced cellular demise (*Zhang et al., 2010*). Studies have demonstrated that inhibition of DJ-1 expression can impact the PTEN pathway, leading to hindered proliferation and induction of apoptosis in PC cells (*Du et al., 2019*). Consequently, it raises the question of whether the anticancer effects of Andro are associated with DJ-1.

Autophagy, an evolutionary conserved catabolic process, experiences robust upregulation in response to cellular stress and nutrient insufficiency. This intricate

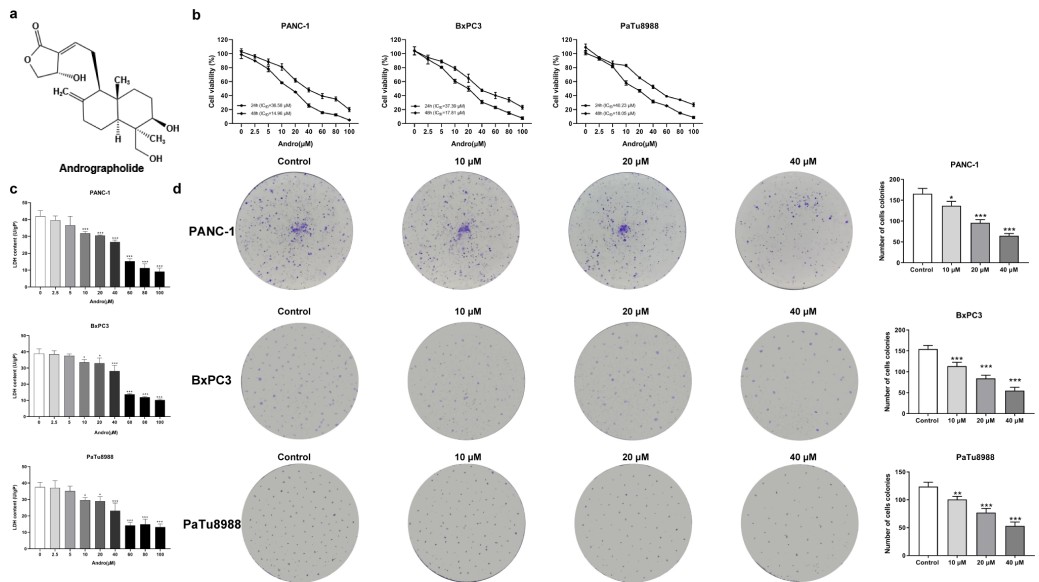

**Figure 1** **Andrographolide inhibits proliferation of pancreatic cancer cells _in vitro_.** (A) The chemical structure of andrographolide. (B) The cell viability of PANC-1, BxPC3 and PaTu8988 groups were detected by CCK-8 assay. These cells were treated with doses of 0, 2.5, 5, 10, 20, 40, 60, 80, 100 $\mu$M Andro for 24, 48 h as determined by CCK-8 assay. The values of IC$_{50}$ in different times were analyzed. (C) Analysis of LDH release in three groups' cells subjected to indicated concentration of Andro. ($*p < 0.05$, $**p < 0.01$, $***p < 0.001$, _vs._ control groups) (D) Cell proliferative ability was analyzed by colony formation assay. ($*p < 0.05$, $***p < 0.001$, _vs._ control groups) ($n = 3$).

mechanism bestows vital sustenance to cellular survival by fostering the recycling of imperative nutrients and averting the buildup of reactive oxygen species (ROS) (_Yuan et al., 2018_). There are now four functional forms of autophagy: cyto-inhibitory, nonprotective, cytotoxic and cytoprotective autophagy (_Gewirtz, 2014_). Among these, cytoprotective autophagy is commonly observed as a response to chemotherapy treatments. In some studies, protective autophagy has been found to reduce apoptosis of cancer cells (_Feng et al., 2018_; _Wang et al., 2018_), as well as the potential prosurvival and protumorigenic role during cancer chemotherapy (_Gewirtz, 2020_), hence the reduction of the therapeutic effect of drugs, indicating that targeting autophagy could be a promising strategy for cancer therapy.

We herein discussed the impact of Andro on PC cells both _in vitro_ and confirmed its effects _in vivo_. Furthermore, we also aimed to investigate the impact of Andro-induced protective autophagy on apoptosis, aiming to elucidate the involvement of DJ-1 in facilitating cell death in PC.

## MATERIALS AND METHODS

### Chemicals and reagents

Andro (A101649, 98%) was obtained from Aladdin (Shanghai, China).

## Cell lines and cell culture

The PANC-1, BxPC3, and PaTu8988 human PC cell lines were procured from iCell Bioscience Inc (Shanghai, China). These cell lines were maintained in DMEM medium (Thermo Fisher Scientific, Waltham, MA, USA), supplemented with 10% FBS (Thermo Fisher Scientific, Waltham, MA, USA). Incubation of the cells was carried out at 37 °C in a 5% $CO_2$ atmosphere.

## CCK-8 assay

The PC cells were seeded in plates and subjected to specific treatments accordingly. Following the varies treatments, 10 µl of CCK-8 solution was introduced to each well, and then the cell lines were incubated in the dark at 37 °C for 1 h. After that, the absorbance at a wavelength of 450 nm was measured using a microplate reader.

## Lactate dehydrogenase release assay

Lactate dehydrogenase (LDH) release assay was applied to detect the cytotoxicity of Andrographis. Briefly, measuring tube, control tube, and standard tube were configured following the manufacturer's protocol. Then, the prepared samples were dispensed into individual wells of a 96-well plate. The value of absorbance was quantified at 450 nm, and the enzyme activity was detected according to the guidelines.

## Colony formation assay

To assess colony formation, cells were seeded in culture plates at a density of $1.5 \times 10^3$ cells per well. The plates were subsequently incubated in a 37 °C incubator for one week to allow colony formation. Following this, the plates were treated with a crystal violet solution and incubated for 30 min to enable staining of the colonies.

## Cell apoptosis assay

To determine apoptosis rates, the Pharmingen Annexin V-FITC Apoptosis Detection Kit I from BD Pharmingen (San Diego, CA, USA) was used following the manufacturer's guidelines. After various treatments, the treated cells were harvested and centrifuged, and then resuspended by PBS at the concentration of $1 \times 10^6$ cells/ml. Subsequently, a mixture of 5 µl of FITC and 5 µl of propidium iodide (PI) was added to each 100 µL of the cell suspension. Following the addition of 400 µl of buffer, the cells were incubated in the dark at 25 °C for 15 min. Finally, the analysis of cell death was conducted using flow cytometry within 1 h (BD Biosciences, San Diego, CA, USA).

## Cell cycle assay

The impact of Andro on the cell cycle was assessed using the Cell Cycle Detection Kit provided by KeyGEN Biotech (Jiangsu, China). After a 24-hour treatment, the cells were collected and centrifuged, followed by PBS washing and adjustment of the cell concentration to $1 \times 10^6$ cells/ml. The resulting single-cell suspension was subjected to another round of centrifugation, and the supernatant was discarded. For fixation, 500 µL of 70% cold ethanol was added to introduced to the cells and incubated at 4 °C for 2 h. Following a PBS rinse, the cells were subjected to staining with 500 µl of PI/RNaseA staining working

solution and maintained in darkness at room temperature for 60 min. Ultimately, the red fluorescence emitted by the stained cells was quantified with 488 nm excitation.

## Immunofluorescence technique

Analysis of LC3 was detected by immunofluorescence method. The cells grown on slides were fixed with 4% paraformaldehyde for a duration of 30 min, followed by washing. The slices were immersed in citrate repair solution and heated in microwave oven until boiling; the sub boiling temperature was kept at 95 °C–98 °C for 10 min. After the wash of slices, the specimens were blocked in blocking buffer for 60 min. After removing the blocking buffer, the diluted primary antibody was applied onto the slides. The slides were subsequently incubated overnight at 4 °C. The specimens were exposed to a fluorescent-labeled secondary antibody, diluted with antibody dilution buffer, in the dark at room temperature for 2 h. Image acquisition was carried out using an inverted fluorescence microscope (Axio vert A1, Carl Zeiss MicroImaging, Jena, Germany).

## Plasmid transfection

The DJ-1 gene sequence (gene sequence number: NM_001123377.2) was synthesized by Shanghai Genechem Company. BamHI and EcoRI enzyme cleavage sites were introduced at both ends of the gene. The DJ-1 synthetic product and pcDNA3.1 plasmid were subjected to double-enzyme digestion, followed by purification. The DJ-1 fragment was ligated into the BamHI site of the pcDNA3.1 plasmid, while the EcoRI site was used for the ligation of the corresponding fragment. The resulting construct was then transformed into chemically competent *E. coli* DH5$\alpha$ cells and cultured overnight at 37 °C. Afterward, the DJ-1-pcDNA3.1 recombinant plasmid was extracted after sequencing identification. After verification, the DJ-1-pcDNA3.1 overexpression plasmid and the pcDNA3.1 control plasmid (vector) were extracted using an endotoxin-free plasmid extraction kit. These purified plasmids were then utilized for subsequent transfection experiments. PANC-1 cells were transfected with DJ-1-pcDNA3.1 overexpression plasmid and vector according to the instructions of Lipofectamine™ 3000, and a blank control group without plasmid was set up. After 8 h, it was changed to fresh complete medium for 24 h. Subsequently, total RNA was extracted from the cells to evaluate the efficiency of overexpression. Following transfection, the cells were detached using trypsin and subsequently seeded into dishes. The cells were then incubated with fresh medium containing the designated doses of Andro for a duration of 48 h before being harvested for the specific experiments.

## Quantitative real-time polymerase chain reaction

Quantitative real-time polymerase chain reaction (qRT-PCR) was performed as described previously (*Xu et al., 2019*). Total mRNA was isolated from transfected cells using Trizol reagent. Reverse transcriptase was used to convert RNA to cDNA (EasyScript® All-in-One First-Strand cDNA Synthesis SuperMix for qPCR, TransGen Biotech, Beijing, China). An amount of 2 μL template cDNA was added to the final volume of 20 μL of reaction mixture.The primer concentration was 500 nM. mRNA expression levels were measured by SYBR green-based quantitative RT-PCR (SYBR Green Master mix; Thermo Scientific, Waltham, MA, USA). Primer sequences were listed in Table 1. The relative amount and

**Table 1  Primer sequences.**

| Primer | Sequence |
| --- | --- |
| GAPDH- F | 5′-TCGGAGTCAACGGATTTGGT-3 |
| GAPDH-R | 5′-TTCCCGTTCTCAGCCTTGAC-3′ |
| DJ-1-F | 5′- GTTCATTTTCAGCCTGGTGTGG-3′ |
| DJ-1-R | 5′- ATGTTATATGTTTACAAGCCTGCAC-3′ |

expression ratio of mRNA were calculated by the $2^{-\Delta\Delta Ct}$ method. The experiment was repeated thrice.

## ROS detection

To determine the production of ROS, the cells were resuspended in a 10 μM DCFH-DA probe solution (S0033M, Beyotime, Shanghai, China) and incubated at 37 °C for 20 min, protected from light. After washing, the cells were detected by CytoFLEX cell fluorometer (Beckman Coulter, Brea, CA, USA). Besides, the DCF fluorescence of cells can be detected with a spectrofluorometer (excitation 488 nm, emission 535 nm).

## *In vivo* experiment design

Forty BALB/c mice were obtained from SPF Biotechnology (Beijing, China) and kept in controlled conditions (temperature: 18–23 °C, humidity: 55–65%). Xenograft models were established by subcutaneously inoculating human pancreatic cancer PANC-1 cells into the right axillary area of the nude mice. These models were subsequently treated with various concentrations (0, 5, 10, 20 mg/kg) of Andro (intraperitoneally) for 4 weeks ($n = 5$). In the experiment aimed at elucidating the involvement of autophagy in Andro's therapeutic effects, twenty mice were categorized into control group, 20 mg/kg Andro group, 3-MA group, and 20 mg/Kg Andro+3-MA group ($n = 5$). A random number table method was used to assign mice. Experimenters were blind to the animal group. Likewise, cage location and the order of treatments were randomly assigned. Upon a duration of four weeks of drug treatment or when the xenografted tumors reached a diameter of 1.5 cm, mice were euthanized with an excess of isoflurane gas. Tumor xenografts and sera were collected from the mice for further evaluation. *In vivo* experiment was conducted with the approval of Ethics Committee of Zhejiang Haikang Biological Products Company (HKSYDWLL2021018).

## Western blot

Western blot was performed as described in a previous study (*Li et al., 2021*) with the following antibodies: Cleaved Caspase-3 (1:1,000), LC3I/II (1:1,000), p62 (1:1,000), and DJ-1 (1:1,000) (Abcam, Cambridge, UK). The internal reference was GAPDH (1:5,000).

## TUNEL assay

The paraffin-embedded tissue sections were subjected to dewaxing using fresh xylene twice, each time for a duration of 10 min. Subsequently, the sections were hydrated using a series of ethanol gradients (100% for 5 min, followed by 90% for 2 min, and 70% for 2 min) and finally distilled water for 2 min. Subsequently, they were treated with protease K (20 μl/ml)
for 30 min at room temperature without DNase. TUNEL analytical solution containing TDT (Beyotime, Shanghai, China) was prepared before using. The slides were incubated in the dark with TUNEL analytical solution at 37 °C for 60 min. After cleaning with PBS and sealing with anti-fluorescence quenching sealing solution, the slide was observed under fluorescence microscope.

### Data analysis and statistics

The experiments were independently repeated a minimum of three times. Data analysis was conducted by SPSS 20.0 (IBM, Armonk, NY, USA) and GraphPad Prism 8.0 (GraphPad Software, San Diego, CA, USA). The results were presented as mean ±standard deviation. Student's $t$-test and one-way analysis of variance were applied to assess group differences. All confidence intervals reported are 95% confidence intervals. Statistical significance was considered when $P < 0.05$.

## RESULT

### Andro inhibits PC cell proliferation and induces apoptosis *in vitro*

The CCK8 assay results demonstrated that Andro exhibited a concentration-dependent and time-dependent reduction in cell viability in various PC cell lines (Fig. 1B). Among the cell lines tested, PANC-1 exhibited the lowest $IC_{50}$ value, indicating its higher sensitivity to Andro. Then the LDH release content indicated the degree of damage to the cell membrane. The results showed that Andro significantly poses damage to various PC cells (Fig. 1C). Consistently, the colony formation assay revealed a significant inhibition of cell proliferation under Andro treatment (Fig. 1D). Flow cytometry analysis confirmed the pro-apoptotic impact of Andro on PANC-1, BxPC3, and PaTu8988 cells (Fig. 2A). Additionally, flow cytometry analysis revealed that Andro treatment resulted in a dose-dependent elevation in the percentage of PC cells in the G2/M phase, indicating cell cycle arrest (Fig. 2B). Immunofluorescence and flow cytometry analysis also demonstrated that Andro treatment increased intracellular ROS levels (Figs. 3A–3C). Overall, our findings indicate that Andro exhibits potent antitumor effects against PC cells *in vitro*.

### Andro suppresses PC cell proliferation and induces apoptosis *in vivo*

In our *in vivo* study, we established xenograft models by subcutaneously inoculating PANC-1 cells into nude mice. Subsequently, the mice were subjected to a 4-week treatment regimen with various concentrations of Andro. Consistent with our *in vitro* findings, Andro exhibited a significant inhibitory effect on PC growth *in vivo*. The greater the concentration of Andro, the slower the growth rate of tumor and the smaller the weight of tumor (Figs. 4A–4C). Besides, it was proved by WB and TUNEL assay that Andro promoted the apoptosis of PC cells *in vivo* (Figs. 4D–4E). This suggests that the antitumor effect of Andro *in vivo* is associated with its ability to induce apoptosis.

### Andro inhibits DJ-1 and up-regulates ROS levels

Andro reduced the expression of DJ-1 protein in a dose-dependent manner (Fig. 5A). We investigated the effect of Andro on cellular ROS levels and examined whether it induces

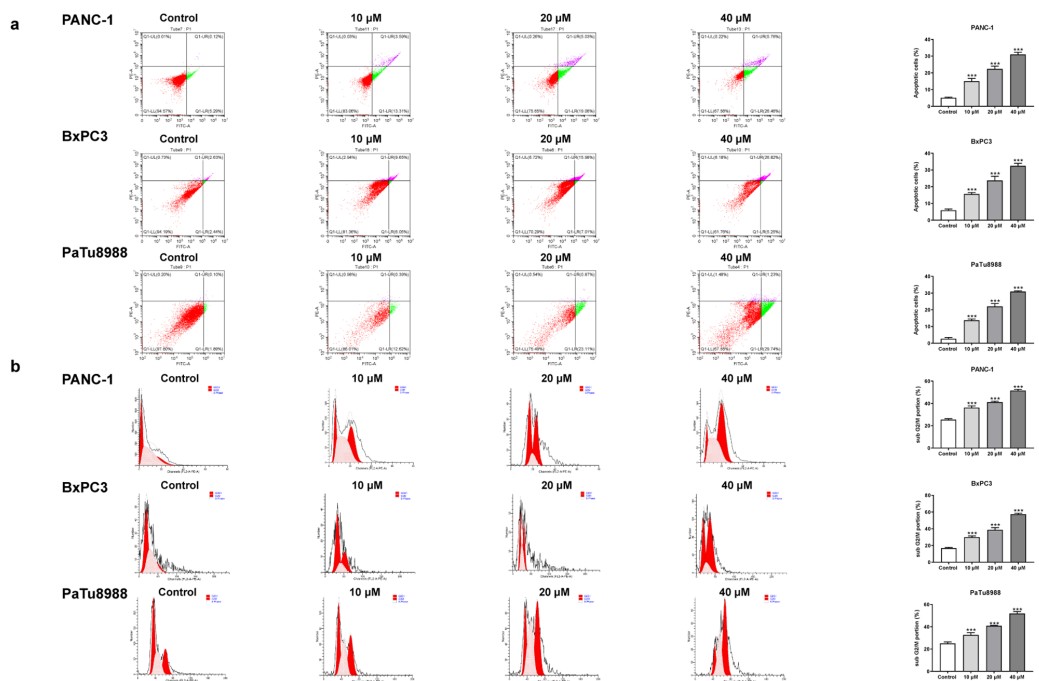

**Figure 2** **Andro induces apoptosis of pancreatic cancer cells *in vitro*.** (A) The cells treated in 0, 10, 20, 40 μM Andro were fixed, the apoptosis rate was measured by flow cytometry. (\*\*\**p* < 0.001, *vs.* control group). (B) The cell cycle progression of pancreatic cancer cells (PANC-1, BxPC3 and PaTu8988) treated with Andro at different concentrations was measured by flow cytometry. (\*\*\**p* < 0.001, *vs.* control group) (*n* = 3).

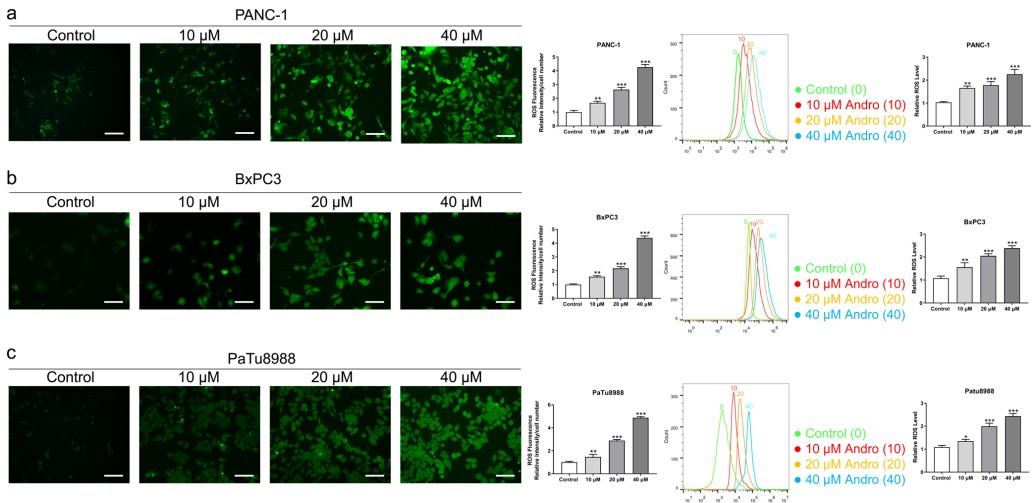

**Figure 3** **Andro increased intracellular ROS levels in a dose-dependent manner.** (A–C) The level of ROS of pancreatic cancer cells (PANC-1, BxPC3 and PaTu8988) treated in 0, 10, 20, 40 μM Andro was evaluated by DCFH-DA assay along with the scale bar of 50 μm, and analyzed by flow cytometry. (\**p* < 0.05, \*\**p* < 0.01, \*\*\**p* < 0.001, *vs.* control group).

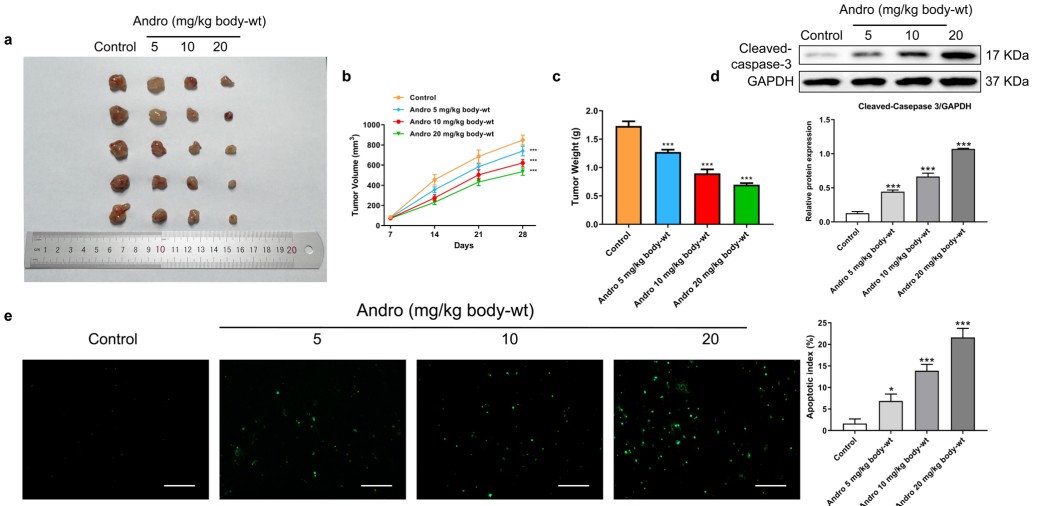

**Figure 4 Andro inhibits pancreatic cancer growth and promotes apoptosis in mice.** Tumor bearing mice were treated in 0, 5, 10, 20 (mg/kg body-wt) Andro for 28 days. (A) The gross features of tumor in four groups, five samples in each group. (B) Tumor volume were measured in 7, 14, 21 and 28 days in four groups and calculated using the formula: Volume = 1/2*length*width$^2$ (C) The mean tumor weight of each group. (D) Western blot depicted alterations in apoptosis-related protein Cleaved-caspase-3 in tumor bearing mice. (***$p < 0.001$ *vs.* control group). (E) The apoptosis rate was measured by TUNEL assay, along with the scale bar of 100 μm. (*$p < 0.05$, ***$p < 0.001$, *vs.* control group).

downregulation of DJ-1, which is known to impair antioxidant response (*Meiser et al., 2016*). The intracellular ROS level markedly increased under Andro treatment, while DJ-1 overexpression reversed the ROS increasing in PC cells (Figs. 5B–5D). Next, we verified the involvement of ROS in the growth inhibition induced by Andro. Firstly, the combination of Andro and ROS scavenger N-acetylcysteine (NAC) could effectively down-regulate the accumulation of ROS caused by Andro (Figs. 5E and 5F). Then CCK-8 experiment (Fig. 5G) and colony cloning experiment (Fig. 5H) were used to prove that the combined treatment of Andro and NAC decreased the growth inhibition induced by Andro. Additionally, the pro-apoptotic effect of Andro was attenuated when NAC was co-administered, as evidenced by WB and flow cytometry analysis (Figs. 5I and 5J). In conclusion, these findings suggested that Andro down-regulated DJ-1 and induced ROS accumulation in cells, leading to enhanced apoptosis.

## Andro induces autophagy in PC cells

Given the established role of ROS as a regulator of autophagy, we aimed to investigate whether Andro regulates autophagy in PC cells. The data demonstrates that Andro treatment dose-dependently increased the accumulation of autophagosomes in PC cells compared to control cells (Fig. 6A). Moreover, Andro treatment resulted in the conversion of LC3-I to lipidated LC3-II, a crucial step in autophagy formation. Additionally, the expression of p62 is significantly reduced upon Andro treatment (Fig. 6B). These data indicated that Andro could induce autophagy in PC cells *in vitro*.

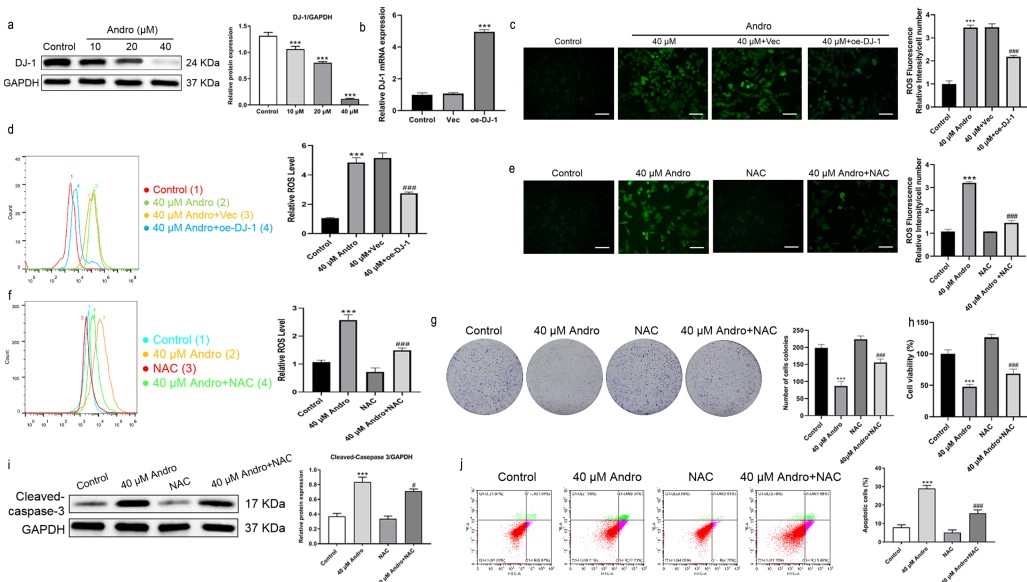

**Figure 5  Andro inhibits DJ-1 and up-regulates ROS levels.** (A) The expression level of DJ-1 in cells treated in indicated concentration of Andro was measured by western blot. (***$p < 0.001$, *vs.* control group) (B). The mRNA expression level of DJ-1 after plasmid transfection was measured by qRT-PCR. DJ-1 mRNA expression of cells respectively treated in control group, Vector group and oe-DJ-1 group was detected by qRT-PCR. (C–D) The level of ROS of pancreatic cancer cells treated in 40 μM Andro, 40 μM Andro + Vec and 40 μM Andro + oe-DJ-1 was evaluated by DCFH-DA assay, along with the scale bar of 50 μm, and analyzed by flow cytometry. (***$p < 0.001$, *vs.* control group, ###$p < 0.001$, *vs.* 40 μM Andro + Vec group) (E–F) The DCFH-DA assay, along with the scale bar of 50 μm, and flow cytometry were applied to detect the level of ROS of four groups including control group, 40 μM Andro group, NAC group and 40 μM Andro + NAC group. (***$p < 0.001$, *vs.* control group, ###$p < 0.001$, *vs.* NAC group). (G) The colony assay was used to detect the inhibitory effect of Andro to NAC. (***$p < 0.001$, *vs.* control group, ###$p < 0.001$, *vs.* NAC group) (H) CCK-8 assay was used to detect the inhibitory effect of Andro to NAC. (***$p < 0.001$, *vs.* control group, ###$p < 0.001$, *vs.* NAC group). (I) The flow cytometry was used to detect the apoptosis effect on pancreatic cancer cells of Andro, NAC and the combination. (***$p < 0.001$, *vs.* control group, ###$p < 0.001$, *vs.* NAC group) (J) Western blot assay was applied to detect the apoptosis effect on pancreatic cancer cells of Andro, NAC and the combination. (***$p < 0.001$, *vs.* control group, #$p < 0.05$, *vs.* NAC group). Abbreviation: NAC, N-acetylcysteine, a ROS scavenger.

## Inhibition of autophagy enhanced the inhibition effect on PC of Andro

To explore the role of autophagy in the inhibitory effect of Andro on PC, we treated PC cells with Andro in combination with 3-MA. Immunofluorescence assay was used to detect LC3. Flow cytometry observed that Andro and 3-MA treatment could increase Andro-induced apoptosis, compared with Andro treatment alone (Fig. 6C). It was found that the 3-MA inhibited autophagy induced by Andro (Fig. 6D). CCK8 assay showed that cell proliferation was reduced in PC cells treated in Andro and 3-MA, compared with treated in Andro (Fig. 6E). Furthermore, we generated xenograft models by inoculating PC cells into nude mice. The combination of 3-MA and Andro significantly enhanced the inhibition effect on PC *in vivo*. This was evidenced by a significant reduction in tumor growth rate, as well as smaller tumor volume and weight compared to the treatment with

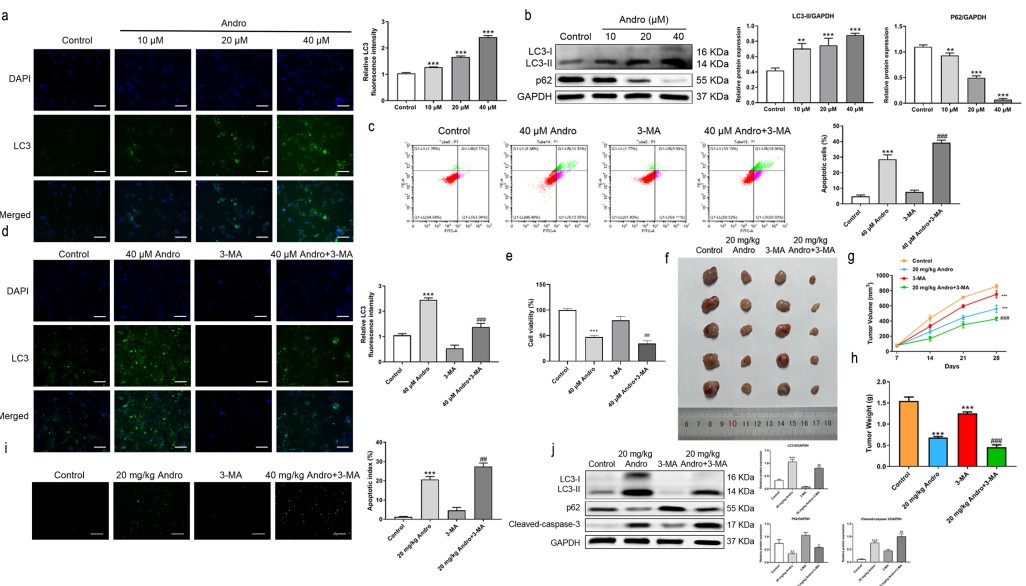

**Figure 6** Andro can induce autophagy of PC cells *in vitro and in vivo*. (A) The immunofluorescence was applied to detect the decrease of LC3-I, along with the scale bar of 50 μm. (***$p < 0.001$, *vs.* control group) (***$p < 0.001$, *vs.* control group) (B) Western blot assay was used to analyze the expression of the autophagy-related proteins including LC3-II, LC3-I and p62. (**$p < 0.01$, ***$p < 0.001$, *vs.* control group). (C) The flow cytometry was applied to analyze the apoptosis effect on pancreatic cancer cells of Andro and 3-MA. (***$p < 0.001$, *vs.* control group, ###$p < 0.001$, *vs.* 3-MA group). (D) The immunofluorescence was used to detect the expression of LC3 in four groups including control group, Andro group, 3-MA group and ANDRO + 3-MA group, along with the scale bar of 50 μm. (***$p < 0.001$, *vs.* control group, ###$p < 0.001$, *vs.* 3-MA group). (E) The CCK-8 assay was used to detect the effect of the combination of Andro and 3-MA on the proliferation. (***$p < 0.001$, *vs.* control group, ##$p < 0.01$, *vs.* 3-MA group). (F) Tumor bearing mice were treated in 0, 20 (mg/kg body-wt) Andro, 3-MA and 3-MA + 20 (mg/kg body-wt) Andro for 28 days. The gross features of tumor in four groups, five samples in each group. (G) The tumor volume was measured in 7, 14, 21 and 28 days in four groups and calculated using the formula: Volume $=1/2*$length$*$width $^2$. (H) The average tumor weight of each group. (***$p < 0.001$, *vs.* control group, ##$p < 0.01$, *vs.* 3-MA group). (I) Effect of Andro and 3-MA on apoptosis of pancreatic cancer cells was detected by TUNEL assay, along with the scale bar of 100 μm. (***$p < 0.001$, *vs.* control group, ##$p < 0.01$, *vs.* 3-MA group). (J) Effect of Andro and 3-MA on apoptosis-related protein and autophagy-related protein of pancreatic cancer cells was analyzed by western blot assay. (***$p < 0.001$, *vs.* control group, #$p < 0.05$, ##$p < 0.01$, *vs.* 3-MA group).

Andro alone (Figs. 6F and 6H). We demonstrated that 3-MA treatment could enhance Andro-induced tumor cell apoptosis through TUNEL assay (Fig. 6I). In addition, the results of WB on autophagy proteins in tumors found that in the compassion with Andro alone, the treatment of Andro and 3-MA weakened the conversion of LC3-I to lipid LC3-II. Additionally, the expression of autophagy-related protein p62 was slightly up-regulated after the combined treatment, and the cleaved-caspase-3 was significantly up-regulated (Fig. 6J). To conclude, our findings suggest that inhibition of autophagy can augment the antitumor effecacy on pancreatic of Andro *in vivo* and *in vitro*. This indicates the potential therapeutic value of combining autophagy inhibitors with Andro in PC treatment.

## DISCUSSION

Andro, a promising drug with strong antitumor effects and low cytotoxicity, has shown potential in treating various types of cancer, including PC (*Deng et al., 2019*; *Lim et al., 2017*). Understanding the mechanisms underlying its effects is crucial in developing effective treatments for PC. In our study, we have shown that Andro exhibits significant inhibitory effects on the growth of PC cells, both *in vitro* and *in vivo*. It achieved this by suppressing cell proliferation and inducing apoptosis. Our findings indicate that Andro treatment results in the downregulation of DJ-1 protein expression. This downregulation induces the ROS accumulation, thereby impeding cell proliferation and promoting apoptosis in PC cells. In the presence of Andro, PC cells were induced to undergo cytoprotective autophagy and exerted protective effects. However, when autophagy was inhibited, the anticancer effect of Andro was significantly enhanced. This highlights the importance of autophagy regulation in the therapeutic potential of Andro. In our study, we reveal for the first time that Andro exerts anti-tumor effects *via* reducing DJ-1 and than promoting ROS accumulation.

Apoptosis is an energy dependent gene programmed cell death mechanism and the main way to eliminate cancer. Some studies have found that Andro has the inhibitory effect in various cancer cells, such as breast cancer (*Peng et al., 2018*), lung cancer (*Luo et al., 2021*), and gastric cancer (*Malat et al., 2021*). In our study, we confirmed that Andro induces apoptosis and inhibits proliferation of PC cells both *in vivo* and *in vitro*. Previous studies have shown that Andro has no significant effect on the growth of normal pancreatic cells (*Bao et al., 2013*). Besides, the significant decrease in DJ-1 expression upon Andro treatment suggests its potential role in the anti-PC effects of Andro. DJ-1 has been recognized as a potential biomarker for predicting the tumor-lymph node-metastasis stage in patients with colorectal cancer. Furthermore, studies have demonstrated that DJ-1 knockdown enhances ROS production in metastatic colorectal adenocarcinoma cells and disrupts mitochondrial accumulation and inhibition of mitochondrial autophagy.

Studies have highlighted the crucial role of DJ-1 in cellular antioxidant defense and the regulation of ROS levels, which play a significant role in tumorigenesis and the progression of cancer (*Olivo et al., 2022*). DJ-1 functions by stimulating the antioxidant system,which helps eliminate ROS and protects cells from cell death caused by oxidative stress. It achieves this by undergoing self-oxidation at Cys106 residues, forming –SO2 or –SO3 modifications . These oxidation states of DJ-1 act as redox sensors, ultimately determining cell fate by activating either autophagy or apoptosis pathways (*Cao et al., 2014*). Consistent with previous findings, our study demonstrated that DJ-1 overexpression led to a notable decrease in ROS levels (*Zhang et al., 2020*). In addition, the down-regulation of DJ-1 induced by Andro led to an increase increase of ROS, which might be the reason for the production of intracellular ROS caused by Andro treatment. The increase in ROS plays a pivotal role in the inhibition of Andro-mediated PC growth. Contradictorily, some studies suggest that inhibiting ROS levels can suppress PC (*Teoh et al., 2007*). This discrepancy arises from the fact that the role of ROS in PC is not always straightforward. Currently, there are two therapeutic strategies targeting ROS. One approach aims to increase ROS to levels toxic to PC cells, which can kill cancer cells without affecting normal cells
(*Gorrini, Harris & Mak, 2013*). Conversely, the opposite strategy involves limiting ROS production and maintaining it at levels that do not promote tumor initiation (*Durand & Storz, 2017*). In our study, using NAC to remove ROS from PC cells could significantly reduce the antineoplastic effect of Andro. Hence, Andro inhibits the growth of PC by suppressing DJ-1, leading to an increase in ROS levels.

Prior review have suggested the complex regulatory relationship between autophagy and ROS in cancer (*Dong et al., 2023*). The role of autophagy manifests diverse characteristics in cancer management. On one aspect, it acts as a cellular guardian, bestowing resistance to therapeutic interventions. On the other hand, it has the capacity to elicit cytotoxic effects and expedite the demise of tumor cells, either autonomously or in synergy with apoptosis (*Zada et al., 2021*). It means that therapeutic agent-induced autophagy can promote not only cell death but also survival, thereby contributing to antitumor efficacy and overcoming drug resistance (*Kumar, Singh & Chaudhary, 2015*). In the majority of instances, autophagy assumes a protective role during cancer treatment, compromising the effectiveness of anticancer drugs. Hence, inhibiting autophagy has been shown to enhance the efficacy of antitumor drugs, highlighting autophagy as a promising focal point for cancer treatment (*Amaravadi, Kimmelman & Debnath, 2019*). In our investigation, we found that Andro treatment increased PC cells autophagic activity. The ROS accumulation induced by Andro may contribute to the activation of cytoprotective autophagy, thereby attenuating the anticancer effects of Andro. When Andro was combined with the autophagy inhibitor 3-MA, autophagy initiation or autophagosome degradation was blocked, leading to a significant reduction in tumor cell proliferation and an augmentation in apoptosis in comparison to sole treatment with Andro. Regrettably, this study does not incorporate a positive control group. Interestingly, previous research has found that Andro exhibits different regulatory effects on autophagy in various cancers, even within the same cancer type, with distinct regulatory mechanisms. Andro has been shown to induce autophagy to suppress the progression of non-small cell lung cancer (NSCLC) or inhibit autophagy to re-sensitize cisplatin-resistant NSCLC cells (*Mi et al., 2016*; *Wang et al., 2022*). However, this study proposes a different mechanism, suggesting that Andro induces protective ROS-dependent autophagy, thereby attenuating Andro's growth inhibitory effect on PC. A similar mechanism has been observed in the inhibition of Hela cell growth by Andro, where researchers inhibited protective autophagy using Taxifolin (*Alzaharna, Alqouqa & Cheung, 2017*).

In conclusion, our study provides experimental evidence supporting the down-regulation of DJ-1 as a key mechanism underlying the antitumor activity of Andro. We have demonstrated that Andro reduces DJ-1 expression, resulting in the buildup of intracellular ROS and subsequent instigation of apoptosis in PC cells. Furthermore, we have shown that Andro also induces cytoprotective autophagy, which weakens its antitumor effect. Our findings propose that directing attention towards cytoprotective autophagy could serve as a potential chemotherapy strategy, and Andro exhibits promising potential as a therapeutic modality for PC. These outcomes enhance our understanding of the

molecular mechanisms implicated in the antitumor efficacy of Andro and provide a basis for further exploration of its therapeutic potential in PC treatment.

### Funding

This work was supported by the Natural Science Foundation of Zhejiang project (Y19H290009). The funders had no role in study design, data collection and analysis, decision to publish, or preparation of the manuscript.

### Grant Disclosures

The following grant information was disclosed by the authors:
Natural Science Foundation of Zhejiang.

### Competing Interests

The authors declare there are no competing interests.

### Author Contributions

- Zhaohong Wang conceived and designed the experiments, performed the experiments, prepared figures and/or tables, authored or reviewed drafts of the article, and approved the final draft.
- Hui Chen performed the experiments, prepared figures and/or tables, and approved the final draft.
- Xufan Cai performed the experiments, analyzed the data, prepared figures and/or tables, and approved the final draft.
- Heqi Bu performed the experiments, analyzed the data, prepared figures and/or tables, and approved the final draft.
- Shengzhang Lin conceived and designed the experiments, authored or reviewed drafts of the article, and approved the final draft.

### Animal Ethics

The following information was supplied relating to ethical approvals (i.e., approving body and any reference numbers):

This study was approved by Ethics Committee of Zhejiang Haikang Biological Products Company.

### Data Availability

The raw measurements are available in the Supplementary Files.

### Supplemental Information

Supplemental information for this article can be found online at http://dx.doi.org/10.7717/peerj.17619#supplemental-information.

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
