# Peer review of "Andrographolide induces protective autophagy and targeting DJ-1 triggers reactive oxygen species-induced cell death in pancreatic cancer"

_PeerJ, doi:10.7717/peerj.17619_

## Round 0.1 · original submission · Major Revisions

Dear authors, thank you for your submission. Your manuscript requires major revisions in order to improve it's scientific quality and soundness. Please, carefully address the reviewers comments. Make sure that data reported matches description and conclusions. Throughout proofreading is advised. And make sure that the materials and methods is clear, precise and complete.

**Language Note:** The Academic Editor has identified that the English language must be improved. PeerJ can provide language editing services - please contact us at [email protected] for pricing (be sure to provide your manuscript number and title). Alternatively, you should make your own arrangements to improve the language quality and provide details in your response letter. – PeerJ Staff

Reviewer 1 ·

Basic reporting

Wang et al. demonstrated that Andrographolide (Andro) inhibited cell proliferation and induced apoptosis in PC cells, both in vitro and in vivo. Moreover, Andro treatment results in the downregulation of DJ-1 protein expression, which play a critical role in Andro-induced ROS and autophagy. Additionally, autophagy functions as a compensatory mechanism to eliminate Andro-induced apoptosis. Manuscript has been nicely written. However, there are some major and minor concerns in the manuscript. In addition, regarding to the conceptual advance of this manuscript, the antitumor effects of Andro have already been documented in various types of cancer context, including PC (Drug Discov Today. 2019, 24(9):1890-1898). Therefore, the novelty of the article is insufficient.

Experimental design

Major points

1. Authors need to incorporate the previous contradictory results between in vitro and in vivo Andro treatment experiments (Toxicol Appl Pharmacol. 2016 Nov 1;310:78-86; Anticancer Drugs. 2017 Oct;28(9):967-976). The literatures showed that Andro suppressed autophagy and subsequently resensitized the resistant cells to cisplatin-mediated apoptosis in lung cancer cells.

2. The production of ROS is considered to be a prosurvival and antiapoptotic factor in PDAC (Clin Cancer Res. 2007 Dec 15; 13(24):7441-50). Rather, in this manuscript, the authors depicted an increase in ROS after Andro treatment. This is contradictory with the well-accepted view. Please discuss this contradiction in detail in the discussion part.

3. In this manuscript, Wang et al. demonstrated that the reduction in DJ-1 expression caused by Andro led to ROS accumulation, subsequently inhibiting the growth of PC cells. In Fig 5, the author did not demonstrate that overexpression of DJ-1 could reverse the growth inhibition and apoptosis caused by Andro. In addition, there is a same problem in Figure 6. The authors need to further investigate the relationship between DJ-1 expression levels and autophagy induced by Andro.

Validity of the findings

no comment

Additional comments

Minor points

1. Line#251-253, the description does not match with the result in Figure 5g and 5h.

2. Some references are too old. The authors are advised to use the most recent five years of references

Reviewer 2 ·

Basic reporting

no comment

Experimental design

no comment

Validity of the findings

no comment

Additional comments

This study investigated the impact of Andro on PC cells both in vitro and confirmed its effects in vivo. They found that Andro treatment resulted in the downregulation of DJ-1 protein expression and the DJ-1 downregulation induced ROS accumulation, thereby impeding cell proliferation and promoting apoptosis in PC cells. They also found that Andro induced cytoprotective autophagy in PC cells and the inhibition of autophagy by inhibitor 3-MA could enhance the apoptosis and the anticancer effect of Andro. However, I still have several concerns as follows:

1. In Figure 3C-E, the order of the pictures does not match the description in the article.

2. The scale bars in the fluorescent image need to be marked.

3. In lines 251-253, the author showed that the combined treatment of Andro and NAC
enhanced the growth inhibition induced by Andro, which seems to be inconsistent with the results in Figure 5G-H.

4. In Figure 1A, the authors determined Andro inhibited the growth of PC cells. Whether Andro has an inhibitory effect on the growth of normal pancreatic cells?

5. The authors demonstrated that Andro decreased the expression of DJ-1. What is the mechanism underlying the downregulation of DJ-1 in Andro-treated PC cells?

6. An important ref regarding autophagy and ROS is suggested to be cited (PMID: 36678588).

---

## Round 0.2 · accepted · Accept

Dear authors, many congratulations. Reviewers were satisfied with your revisions, and I do agree that the additional experiment requested may be out of scope this time. Your manuscript has been now accepted for publication.

Reviewer 1 ·

Basic reporting

no comment

Experimental design

no comment

Validity of the findings

no comment

Additional comments

The authors have addressed most of the points raised by the reviewers. However, I still suggest that the author should increase the overexpression experiments of DJ-1 to clarify the relationship between DJ-1 expression levels and autophagy induced by Andro.